# Identification of early predictors and model for bacterial infection in diabetic ketoacidosis patients: A retrospective study

**Yaping Hao**[☯], **Lei Yang**[☯], **Xiaomei Meng, Yuxiao Tang, Liang Wang**[iD]*

Department of Endocrinology, Affiliated Yantai Yuhuangding Hospital of Qingdao University, Yantai, Shandong, China

☯ These authors contributed equally to this work.

* laiyangwangliang@163.com

## Abstract

### Purpose

The purpose of this report was to identify effective indicators capable of predicting bacterial infection during the early stages of diabetic ketoacidosis (DKA) and to establish a diagnostic model suitable for clinical application.

### Methods

This was a retrospective cross-sectional study. Between February 2018 and May 2023, Yuhuangding Hospital admitted 101 DKA patients, of whom 45 were diagnosed with bacterial infections. A confirmed bacterial infection was defined as documented bacteriological evidence in any bacterial sample. Clinical parameters and biological markers (including cortisol, C-reactive protein (CRP), procalcitonin, etc.) were recorded during the initial DKA phase. Multivariate regression analysis was employed to construct a diagnostic model.

### Results

CRP (OR = 1.014, 95% CI: 1.002–1.026, $p$ = 0.017) and cortisol (OR = 1.007, 95% CI: 1.002–1.012, $p$ = 0.003) were found to have an independent association with bacterial infection in DKA patients. The area under the receiver operating characteristic curve (AUC) for CRP in identifying bacterial infection was 0.855 (95% CI, 0.771–0.917), with a sensitivity of 76.1% and a specificity of 83.6%. The AUC for cortisol in identifying bacterial infection was 0.847 (95% CI, 0.761–0.911), with a sensitivity of 71.7% and a specificity of 89.1%. A joint diagnostic model based on cortisol and CRP was developed through multifactor regression analysis. The AUC of this diagnostic model was 0.930 (95% CI, 0.862–0.972), resulting in a sensitivity of 93.5% and a specificity of 80.0%.

### Conclusion

CRP and cortisol are early indicators of bacterial infection in DKA patients. Furthermore, based on their combination, the regression diagnostic model exhibits enhanced diagnostic performance.

**Data Availability Statement:** The datasets generated and analyzed during the current study are available in supplementary data.

**Funding:** The author(s) received no specific funding for this work.

**Competing interests:** The authors have declared that no competing interests exist.

**Abbreviations:** DKA, Diabetic ketoacidosis; CRP C, reactive protein; IL-6, Interleukin-6; ACTH, Adrenocorticotropic hormone; ROC, Receiver operating characteristic; AUC, Area under the ROC curve; HPA, Hypothalamic-pituitary-adrenal.

## Introduction

Diabetic ketoacidosis (DKA) represents a potentially life-threatening complication of diabetes mellitus, characterized by hyperglycemia, ketosis, and metabolic acidosis. It accounts for 4–9% of all hospital discharges among patients with diabetes mellitus as the primary cause for their acute hospital admission, and recent studies have reported a mortality rate ranging from approximately 2 to 5%. In DKA, infection has become the primary contributing factor to illness and mortality [1]. Among the pathogenic microorganisms encountered, *Escherichia coli* and *Klebsiella pneumonia* are frequently implicated [2].

The early detection of bacterial infections and the timely administration of appropriate antibiotic treatments are pivotal in enhancing patient outcomes. However, clinicians often face challenges distinguishing between infected and non-infected individuals due to DKA's overlapping clinical symptoms with other infections, such as leukocytosis and elevated body temperature. Although bacteremia can be definitively confirmed through blood culture, it exhibits a lag in detection. Consequently, scientists have long sought to identify early predictors of occult bacterial infection in DKA patients. Previous research has investigated the potential predictive value of factors including age, gender, body temperature, blood glucose levels, pH, bicarbonate levels, total leukocyte count, and total neutrophil count [3–6], which possess limited value. However, these variables have demonstrated limited utility in predicting bacterial infection in DKA.

Procalcitonin, frequently used as an indicator of bacterial infection, has recently been associated with nonspecific elevations in cases of diabetic ketoacidosis [7, 8]. The precise underlying mechanism for this association remains unknown. Notably, a significant correlation has been observed between bacterial infection in DKA and the biomarkers C-reactive protein (CRP) and interleukin-6 (IL-6). However, the extent to which these biomarkers can be effectively employed to predict bacterial infection remains uninvestigated [3, 9, 10].

Blood cortisol levels are significantly elevated under a variety of stress conditions, including sepsis, surgery, and burns, and this elevation is positively correlated with the severity of the stress or disease [11–14]. However, as far as we know, no study has explored cortisol as a potential predictor for bacterial infection in DKA. Considering the potential exacerbation of DKA by bacterial infection, herein, we hypothesized that blood cortisol levels could be notably higher in DKA patients with bacterial infection.

This study examined widely used indicators documented in previous studies to evaluate bacterial infection in DKA. Furthermore, potential predictors of bacterial infection, including blood cortisol, adrenocorticotropic hormone (ACTH), and procalcitonin, were assessed. We performed multivariable logistic regression analyses to identify factors associated with bacterial infection of DKA patients.

## Materials and methods

### Patient selection

This retrospective cross-sectional study involved a cohort of 242 patients diagnosed with DKA who visited the Endocrinology Department of Yantai Yuhuangding Hospital from February 2018 to May 2023. Among these patients, 141 individuals were excluded from the study for the following reasons: 1) age under 18 and over 75 years; 2) with infection of unknown etiology; 3) received drugs that affect the hypothalamic-pituitary-adrenal (HPA) axis, specifically glucocorticoids, anesthetic drugs, and psychotropic medications; 4) unknown endocrine, liver, kidney or central nervous system dysfunction. Finally, a total of 101 subjects were recruited and utilized for the implementation of this study. Among these

participants, 46 had bacterial infections, while 55 did not. Bacterial infection was diagnosed by examining blood or body fluid cultures or considering a combination of clinical symptoms, biochemical markers, and imaging findings. Patients with DKA who did not have a bacterial infection (control group) and suspected evidence of viral infection included those without any infection. Latest data access for this research purpose was 1[st] January 2024. This study protocol has been approved by the Ethical Committee of Yantai Yuhuangding Hospital in China (No. 2023–418). The study was performed in accordance with the ethical standards as laid down in the 1964 Declaration of Helsinki and its later amendments or comparable ethical standards.

## Measurements of clinical and laboratory indicators

Upon admission, patient characteristics, including age, sex, and body temperature, were meticulously recorded. The determination of diabetes type was made either at the time of discharge or during subsequent follow-up visits.

To assess cortisol and ACTH levels, blood samples were collected at 8 AM on the day following admission, and electro-chemiluminescence (ECL) was employed for analysis (Cobas e601; Roche, Switzerland). Additional blood samples were obtained before initiating DKA treatment to evaluate plasma glucose, serum bicarbonate, lactate, total leukocyte count, neutrophil count, CRP, and procalcitonin levels. Plasma glucose and serum bicarbonate concentrations were analyzed using the AU5800 automated chemistry analyzer (Beckman Coulter, Brea, CA, USA). Lactate levels were determined using a blood-gas analyzer (ABL800; Radiometer, Denmark). Furthermore, total leukocyte and neutrophil counts were measured using automatic blood counting systems (XN2000; SYSMEX, Japan). CRP levels were assessed using Latex-enhanced scattering nephelometry (BC5390; Mindray, China), while procalcitonin levels were determined through ECL immunoassay (Cobas e411; Roche, Switzerland).

## Statistical analysis

Before statistical analysis, the data underwent initial processing procedures, including eliminating outliers and interpolating missing values. Multiple imputation techniques were used to handle missing values. Descriptive statistics were employed to summarize continuous variables. Values with mean ± standard deviation (SD) were stated for variables following a normal distribution, while the median (first and third quartile) was utilized for variables not conforming to a normal distribution. An independent sample t-test was used for normally distributed data, while a rank sum test was used for non-normally distributed data, and a Chi-square test was conducted for categorical data. Binary logistic regression was conducted to examine the relationship between predictor variables and a binary outcome variable.

Variables that showed significant changes in the univariate analysis were included in addition to the "neutrophil" variable, which was excluded due to severe multicollinearity. A clinical prediction model was developed using logistic regression analysis, incorporating CRP and cortisol as parameters. Receiver operating characteristic (ROC) curve analysis was used to evaluate the performance of CRP, cortisol, and the combined prediction model regarding diagnosis. The Normal Z test was used to compare the area under the ROC curve (AUC) values of each predictor or model. All statistical analyses, except for ROC curve analysis using MedCalc version 19.0 (MedCalc Software Ltd, Ostend, Belgium), were conducted using SPSS version 24.0 (SPSS, Inc., Chicago, IL, USA). Statistical significance was determined at a p-value of less than 0.05.

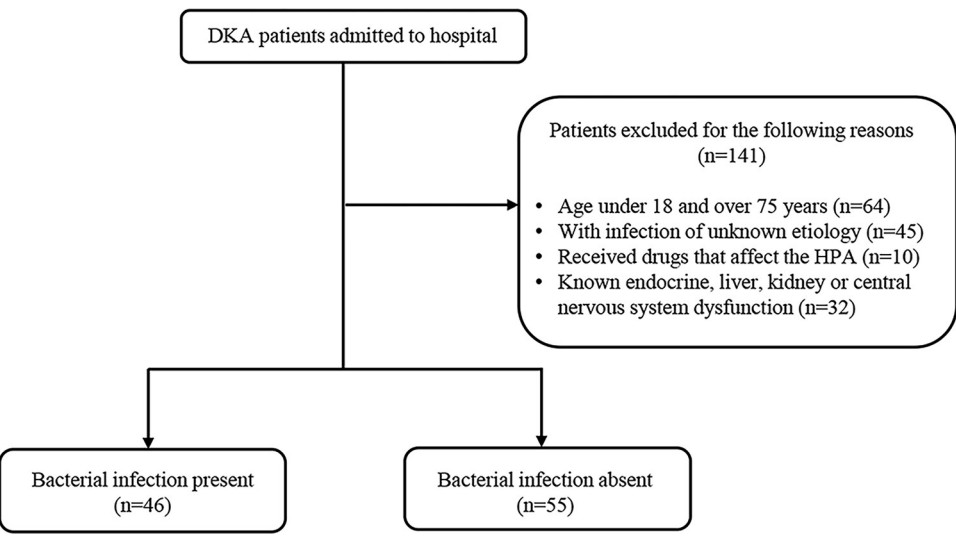

**Fig 1. Flowchart of the sample selection.** Study flowchart depicting patient eligibility, reasons for exclusion, and final group allocation in the study.

## Results

### Characteristics and comparison of bacterial infection group and non-bacterial infection group in DKA patients

This study enrolled a cohort of 101 DKA patients, of which 46 received a bacterial infection diagnosis, as illustrated in **Fig 1**. **Table 1** provides an overview of the demographic and clinical characteristics of these patients. Notably, 18 out of the 46 patients with bacterial infection were directly diagnosed through blood culture or body fluid culture, while the remaining cases relied on clinical manifestations, biochemical markers, and imaging evidence for diagnosis. There were no statistically significant differences in gender, body temperature, blood ACTH, or lactic acid levels between the patients with and without bacterial infections ($p > 0.05$). In

**Table 1. Clinical and biochemical features of DKA participants.**

| Variable | Bacterial infections (n = 46) | Non-bacterial infections (n = 55) | t/z/$\chi^2$ | P |
|---|---|---|---|---|
| Age (years) | 48.63 ± 16.50 | 39.38 ± 15.69 | 2.882 | 0.005 |
| Male/female (n/n) | 23/23 | 31/24 | 0.408 | 0.523 |
| Diabetic type (1/2) | 7/39 | 18/37 | 4.123 | 0.042 |
| Temperature (˚C) | 36.52 ± 0.75 | 36.46 ± 0.47 | 0.512 | 0.61 |
| Glucose (mmol/L) | 22.96 ± 6.51 | 26.90 ± 7.68 | -2.751 | 0.007 |
| Cortisol (μg/dL) | 27.73 ± 9.61 | 16.61 ± 6.54 | 6.662 | < 0.001 |
| ACTH (pg/mL) | 12.73 (6.74, 18.88) | 9.94 (5.00, 17.41) | 1.343 | 0.179 |
| Lactic acid (mmol/L) | 1.67 ± 0.50 | 1.64 ± 0.66 | 0.289 | 0.774 |
| HCO$_3$ (mEq/L) | 5.60 (3.88, 8.93) | 9.40 (5.90, 13.30) | 3.291 | 0.001 |
| Leukocyte (×10$^3$/μL) | 18.08 (11.40, 24.55) | 12.89 (9.38, 18.47) | 2.977 | 0.003 |
| Neutrophil (×10$^3$/μL) | 15.34 (10.53, 20.97) | 10.58 (7.26, 15.06) | 3.45 | 0.001 |
| C-reactive protein (mg/L) | 160.55 (70.23, 264.16) | 11.81 (3.53, 49.75) | 6.13 | < 0.001 |
| Procalcitonin (μg/L) | 3.10 (0.95, 4.89) | 0.52 (0.13, 1.51) | 4.876 | < 0.001 |

Values are mean ± standard deviation (SD). P values represent the comparisons between the two groups. Abbreviation: ACTH, adrenocorticotropic hormone.

**Table 2. Multivariate regression analysis of potential predictors of bacterial infection in DKA patients.**

| Variable | B | SE | Wals | P | OR | 95%OR | |
|---|---|---|---|---|---|---|---|
| | | | | | | Lower limit | Upper limit |
| Age (years) | 0.036 | 0.023 | 2.314 | 0.128 | 1.036 | 0.99 | 1.**085** |
| Diabetic type | 0.197 | 0.959 | 0.042 | 0.837 | 1.217 | 0.186 | 7.981 |
| Glucose (mmol/L) | -0.079 | 0.058 | 1.896 | 0.168 | 0.924 | 0.825 | 1.034 |
| Cortisol (µg/dL) | 0.007 | 0.002 | 8.947 | 0.003 | 1.007 | 1.002 | 1.012 |
| HCO$_3$ (mEq/L) | -0.035 | 0.125 | 0.077 | 0.781 | 0.966 | 0.757 | 1.233 |
| Leukocyte (×10$^3$/µL) | 0.043 | 0.077 | 0.315 | 0.575 | 1.044 | 0.898 | 1.215 |
| C-reactive protein (mg/L) | 0.014 | 0.006 | 5.648 | 0.017 | 1.014 | 1.002 | 1.026 |
| Procalcitonin (µg/L) | 0.369 | 0.233 | 2.513 | 0.113 | 1.446 | 0.916 | 2.281 |
| Constant | -6.338 | 3.029 | 4.378 | 0.036 | 0.002 | | |

B, Partial regression coefficient values; SE, Standard error; Wals, Wald chi-square value; P, probability; OR, odds ratio

contrast, when compared to the non-bacterial infection group, the patients with bacterial infections exhibited higher ages, a greater percentage of individuals diagnosed with type 2 diabetes, increased leukocyte and neutrophil counts, elevated blood cortisol levels, and higher levels of CRP and procalcitonin levels. Conversely, the bacterial infection group displayed lower blood glucose and serum bicarbonate levels ($p<0.05$).

## Establishing a diagnostic model for DKA with bacterial infection

Univariate analysis identified nine potentially significant indicators: age, diabetes type, blood glucose, cortisol, serum bicarbonate, leukocyte, neutrophil, CRP, and procalcitonin ($p<0.05$, **Table 1**). In the subsequent multivariate logistic regression analysis, the independent variable neutrophil was excluded due to multicollinearity with other variables. The analysis revealed that blood cortisol (OR = 1.007, 95% CI: 1.002–1.012, $p = 0.003$) and CRP (OR = 1.014, 95% CI: 1.002–1.026, $p = 0.017$) were two independent predictors of bacterial infection in DKA (**Table 2**). A combined linear prediction model incorporating both variables was developed, expressed by the formula: Logit P = -6.596 + (0.008 × cortisol) + (0.020 × CRP) (**Table 3**).

## Diagnostic value of cortisol, CRP, and their combined diagnostic model in identifying DKA with a bacterial infection

To assess the predictive value of blood cortisol and CRP levels, as well as their combined diagnostic model, for bacterial infection in DKA, ROC curve analysis (**Table 4** and **Fig 2**) was conducted. The findings indicated that CRP had an AUC of 0.855 (95% CI, 0.771–0.917) in distinguishing bacterial infection, with an optimal cut-off value of 71.53 mg/L. The sensitivity, specificity, positive predictive value, negative predictive value, and diagnostic accuracy were

**Table 3. Construction of a diagnostic model for DKA with bacterial infection.**

| Variable | B | SE | Wals | P | OR | 95%OR | |
|---|---|---|---|---|---|---|---|
| | | | | | | Lower limit | Upper limit |
| Cortisol (µg/dL) | 0.008 | 0.002 | 16.897 | < 0.001 | 1.008 | 1.004 | 1.012 |
| C-reactive protein (mg/L) | 0.02 | 0.004 | 19.844 | < 0.001 | 1.02 | 1.011 | 1.029 |
| Constant | -6.596 | 1.391 | 22.48 | < 0.001 | 0.001 | | |

Predictive formula: Logit P = -6.596+(0.008×Cortisol) + (0.020×C-reactive protein)

**Table 4. Diagnostic value of cortisol, CRP and their combined Logit P diagnostic model for bacterial infection.**

| Index | Cut off | Sensitivity | Specificity | PPV | NPV | Accuracy | AUC |
|---|---|---|---|---|---|---|---|
| | | (%) | (%) | (%) | (%) | (%) | (95%CI) |
| Cortisol (µg/dL) | 23.33 | 71.7 | 89.1 | 84.6 | 79 | 81.2 | 0.847*# |
| | | | | | | | (0.761–0.911) |
| C-reactive protein (mg/L) | 71.53 | 76.1 | 83.6 | 79.6 | 80.7 | 80.2 | 0.855*# |
| | | | | | | | (0.771–0.917) |
| Combined Logit P model | -0.93 | 93.5 | 80 | 79.6 | 93.6 | 86.1 | 0.93 |
| | | | | | | | (0.862–0.972) |

*p<0.05 Comparison to combined Logit P model

#p>0.05 Comparison between cortisol and C-reactive protein

PPV, positive predictive value; NPV, negative predictive value; AUC, area under the ROC curve; CI, confidence interval

76.1, 83.6, 79.6, 80.7, and 80.2%, respectively. Similarly, blood cortisol exhibited an AUC of 0.847 (95% CI, 0.761–0.911) in identifying bacterial infection, using a cut-off value of 23.33 ug/ dL with the sensitivity, specificity, positive predictive value, negative predictive values, and

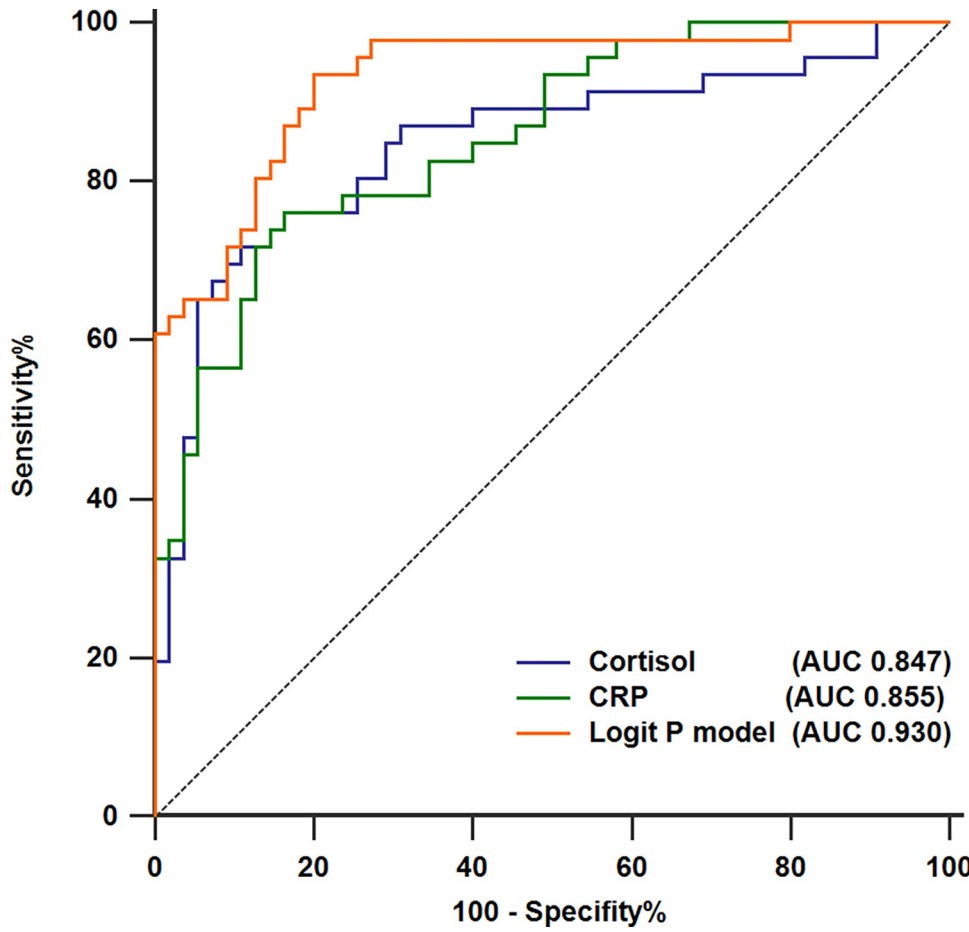

**Fig 2. ROC curves: Cortisol, CRP, and combined model for bacterial infection in DKA.** ROC curves demonstrating the predictive performance of blood cortisol (AUC 0.847), CRP (AUC 0.855), and their combined Logit P model (AUC 0.930) for bacterial infection in diabetic ketoacidosis.

diagnostic accuracy of 71.7, 89.1, 84.6, 79.0, and 81.2%, respectively. The predictive value of cortisol was comparable to that of CRP in predicting bacterial infection. Moreover, their combined Logit P diagnostic model demonstrated an AUC value of 0.930 (95% CI, 0.862–0.972), with a cut-off value of -0.93. The sensitivity, specificity, positive predictive value, negative predictive value, and diagnostic accuracy were determined to be 93.5, 80.0, 79.6, 93.6, and 86.1%, respectively. The model showed a significantly higher diagnostic value than cortisol or CRP alone.

## Discussion

Detecting bacterial infection in the early stages of DKA has consistently presented a formidable challenge for clinicians. This study has reaffirmed the findings established in prior research. Specifically, it was observed that elevated CRP levels exhibited a significant correlation with bacterial infection, whereas factors such as age, gender, body temperature, blood glucose, bicarbonate levels, total leukocyte count, and neutrophil count showed no significant correlation with bacterial infection during the early stage of DKA. Moreover, a noteworthy correlation emerged between the overall concentration of blood cortisol and the occurrence of bacterial infections. However, there was no discernible correlation between bacterial infections and the concentrations of ACTH and procalcitonin. We conducted ROC curve analysis to evaluate the diagnostic efficacy of CRP and blood cortisol levels in distinguishing between bacterial and non-bacterial infections during the early stages of DKA.

The results suggest that CRP and blood cortisol levels can effectively predict bacterial infection with equivalent diagnostic efficacy. Using logistic regression, we constructed a combined logit P diagnostic model based on cortisol and CRP, demonstrating a diagnostic sensitivity of 93.5% and specificity of 80%, with an optimal cut-off value of -0.93. The diagnostic value of this model surpassed that of CRP or cortisol alone. It is anticipated to be an effective tool for early identification of bacterial infection in DKA in clinical settings.

Patients with DKA may present with a range of nonspecific symptoms and signs, including headache, shortness of breath, nausea, vomiting, abdominal pain, altered mental status, fever, and tachycardia. These clinical manifestations are common in both DKA and bacterial infections, making it challenging to identify co-existing bacterial infections in DKA patients. Notably, most DKA patients have normal or low body temperature, even in the presence of bacterial infection [1]. This sometimes leads clinicians to overlook the possibility of bacterial infection. While increased leukocyte counts are often observed in DKA, this is more likely due to stress or hemoconcentration [6]. Numerous studies have demonstrated that the total leukocyte counts in DKA did not significantly correlate with bacterial infection; instead, they reflect the severity of DKA [4–6]. One study suggested that a count of band neutrophils exceeding 10% could effectively predict recessive bacterial infections in adult DKA patients, with a sensitivity of 100% and a specificity of 80% [5]. However, the practical application of this finding is limited as most laboratories do not report counts of band neutrophils separately.

Procalcitonin, a frequently used diagnostic marker for sepsis, may exhibit nonspecific elevations in burns, surgery, and pancreatitis. Recent studies have also indicated that nonspecific increases in procalcitonin levels among patients with DKA might be attributable to elevated levels of particular inflammatory factors, although the precise mechanism remains unknown [8, 15, 16]. Herein, our results observed that body temperature, white blood cell count, neutrophil count, and procalcitonin level did not significantly correlate with bacterial infection in DKA.

CRP has a lower specificity than procalcitonin and can be elevated in bacterial or viral infections and in cases of non-infective inflammatory conditions, including DKA [17, 18].

However, a significant correlation has been established in most studies between CRP levels and the incidence of bacterial infections in patients with DKA. This might be attributed to the increased IL-6 production during bacterial infection, which caused an upsurge in hepatic CRP synthesis [3, 9, 10]. IL-6 and CRP levels were concurrently elevated and significantly elevated during the initial phase of DKA with sepsis, followed by a significant decrease after DKA improvement. This indirectly validated this mechanism [10]. However, the predictive value of CRP and IL-6 in differentiating bacterial infections among patients with DKA has yet to be investigated. CRP was an accurate predictor of early bacterial infection in DKA. According to the findings in our study, CRP was an effective predictor of early bacterial infection in DKA, with a cut-off value of 71.53 mg/L, a sensitivity of 76.1%, and a specificity of 83.6%. CRP level typically peaks approximately 36–50 hours after inflammatory stimulation [19], resulting in a diagnostic delay. As a result, we must monitor dynamic changes in CRP levels during the early stages of DKA to predict peaks, as performed in our study. However, we did not measure IL-6 levels in this study. Nevertheless, considering the significant correlation between IL-6 and CRP levels reported in previous studies, IL-6 will be omitted from subsequent logit P diagnostic models, notwithstanding its detection.

Cortisol, a vital stress hormone, rapidly increases the bloodstream levels during periods of severe stress. The prominent surge in cortisol levels performs an essential function in maintaining bodily survival by initiating diverse physiological responses, such as blood pressure elevation, inflammation inhibition, and immune response compromise [20]. Blood cortisol levels positively correlate with the severity of stress conditions such as sepsis, surgery, and burns. Interestingly, 'ACTH-cortisol dissociation' occurs in most cases, wherein blood ACTH and cortisol levels do not increase concurrently, and occasionally, ACTH levels even decrease [13, 21–23]. It is hypothesized that this could be related to the adrenal gland being stimulated by peripheral factors (e.g., cytokines and catecholamines) other than ACTH and reduced cortisol clearance [21, 22, 24]. Our study uncovered the occurrence of 'ACTH-cortisol dissociation' in DKA, regardless of bacterial infection. This implies that the elevation of blood cortisol during DKA may not be attributable to HPA axis activation but to this mechanism. Nonetheless, our findings indicated that blood cortisol levels, either alone or in conjunction with CRP, could be a reliable predictor of early bacterial infection in DKA, as opposed to ACTH.

Several limitations should be noted in our study. Firstly, it is essential to acknowledge the absence of a unified diagnostic standard for bacterial infections, particularly since blood cultures and bodily fluids are unreliable microbiological diagnostic tools. This might potentially distort the true incidence of bacterial infections. In order to ensure the reliability of our sample, we included patients with both bacterial and non-bacterial infections in DKA with great care, relying on a comprehensive clinical diagnosis. Secondly, our study had a limited sample size. More clinical cases are needed to verify the diagnostic performance of blood CRP, cortisol levels, and their combined logit P model. However, the number of independent variables included in the binary logistic regression analysis incorporated in the sample size ensures the successful construction of the logit P prediction model. Thirdly, to maintain the blood cortisol levels, we eliminated interfering factors that could affect cortisol levels and restricted the age of selected patients. This, nevertheless, limited the clinical application of the blood CRP, cortisol levels, and logit P model screened in this study. Consequently, further investigation in a larger clinical population is still necessary to determine the actual utility of these indicators or models. Besides, we did not assess the differences in the independent variables between bacterial and viral infections. Viral infections are frequently challenging to diagnose, and in contrast to bacterial infections, there is typically no specific treatment or urgent need for discovery. Therefore, we categorized patients with suspected viral infections and no infections into the non-

bacterial infection group. Although this methodology might make it more challenging to identify bacterial infections, it was more in line with contemporary clinical practice.

## Conclusions

Our study has identified cortisol and CRP as valuable markers that can aid in the early detection of bacterial infections in DKA patients. Leveraging these two predictors, we have developed an integrated predictive model. Using predictive markers or models is pivotal in swiftly identifying concurrent bacterial infections in DKA patients. This, in turn, enables the timely initiation of antibiotic therapy and ultimately enhances the prognosis of DKA.

## Supporting information

**S1 Data.**
(XLSX)

## Acknowledgments

We thank Medjaden Inc. for its assistance in the preparation of this manuscript.

## Author Contributions

**Conceptualization:** Liang Wang.

**Data curation:** Yaping Hao, Lei Yang, Xiaomei Meng, Yuxiao Tang, Liang Wang.

**Formal analysis:** Yaping Hao, Lei Yang, Xiaomei Meng, Liang Wang.

**Resources:** Yuxiao Tang.

**Writing – original draft:** Yaping Hao, Lei Yang.

**Writing – review & editing:** Yaping Hao, Lei Yang, Xiaomei Meng, Yuxiao Tang, Liang Wang.

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
