## [Decision Letter · Decision Letter 0]

25 Sep 2024

PONE-D-24-30892Identification of early predictors and model for bacterial infection in diabetic ketoacidosis patients: A retrospective studyPLOS ONE

Dear Dr. Wang,

Thank you for submitting your manuscript to PLOS ONE. After careful consideration, we feel that it has merit but does not fully meet PLOS ONE’s publication criteria as it currently stands. Therefore, we invite you to submit a revised version of the manuscript that addresses the points raised during the review process.

We look forward to receiving your revised manuscript.

Kind regards,

Timotius Ivan Hariyanto, M.D.

Academic Editor

PLOS ONE

Journal Requirements:

3. In the online submission form, you indicated that the datasets generated and analyzed during the current study are available from the corresponding author on reasonable request.. 

Reviewers' comments:

Reviewer's Responses to Questions

**Comments to the Author**

1. Is the manuscript technically sound, and do the data support the conclusions?

Reviewer #1: Yes

Reviewer #2: Yes

2. Has the statistical analysis been performed appropriately and rigorously? 

Reviewer #1: Yes

Reviewer #2: Yes

3. Have the authors made all data underlying the findings in their manuscript fully available?

Reviewer #1: Yes

Reviewer #2: Yes

4. Is the manuscript presented in an intelligible fashion and written in standard English?

Reviewer #1: Yes

Reviewer #2: Yes

5. Review Comments to the Author

Reviewer #1: • The manuscript describes the technically sound scientific research with data supporting the conclusions. Appropriate controls are available. The statistical analysis is appropriate. The conclusions are drawn appropriately based on the whole data presented. The manuscript is presented in an intelligible fashion and in standard English.

• Title: The words of the title are clear, not long, reflects study variables and doesn’t contain junk information.

• Abstract: It is clear, includes the research problem. objectives. the research subjects & methods, results, conclusion, keywords and in separate page. But, the type of the study is not mentioned i.e. case control study. But description of the type of the study should be mentioned.

• The list of tables and figures are clear and well-formulated.

• Introduction: Provide necessary context, clearly specified the focus of the research, and show the relevance and importance of the research topic. It clearly stats the problem or question for research addresses. It outlines the specific objectives of the research. But please revise line 43 is it 4-6% of all hospital discharge or admissions? And lines 44 and 45 for the wording, mortality rate ranging from approximately 2 to 5%.

• Subjects and methods: It describes the research method, research population, sample, tool and the statistical methods. But description of the type of the study should be mentioned. But please revise line 83 and figure 1 P21 for the exclusion criteria 1) age under 18 an over 75? Line 99 please revise the methodology of Cobas e601, Roche, Switzerland. It is electrochemiluminescence (ECL) technology for immunoassay analysis.

• Results: Answers the research questions, verify/check the hypotheses and proper statistical methods.t

• Discussion: Results are discussed and compared with previous studies.

• Summary, conclusion, recommendations: It gives summary and conclusion with mentioning the recommendations and suggestions.

Finally, my recommendation is to accept with minor changes.

Reviewer #2: A well conceptualised study with good methodology.

Manuscript is well-written.

However, there are severe limitations especially the limited sample size.

Authors must specify the medications that affect the HPA axis in line 84

6. PLOS authors have the option to publish the peer review history of their article (what does this mean?). If published, this will include your full peer review and any attached files.

Reviewer #1: **Yes: **Samia A Girgis, Ain Shams University, Faculty of Medicine, Cairo, Egypt

Reviewer #2: No

---

## [Author Response · Author response to Decision Letter 0]

6 Nov 2024

Dear Dr. Timotius Ivan Hariyanto,

Thank you very much for your decision letter and advice on our manuscript (PONE-D-24-30892) entitled “Identification of early predictors and model for bacterial infection in diabetic ketoacidosis patients: A retrospective study”. We have revised the manuscript accordingly, and all amendments are indicated by red font in the revised manuscript. In addition, our point-by-point responses to the comments are listed below this letter.

Replies To Reviewer 1

1. Abstract: The type of the study is not mentioned i.e. case control study. But description of the type of the study should be mentioned.

Response: Thank you for your insightful suggestion. This study is a clinical prediction model study based on retrospective cross-sectional data, as we used historical data collected at a specific time point to develop our model. This point has been briefly mentioned in the Abstract of the revised manuscript (Page 2, Lines 22).

2. Introduction: Please revise line 43 is it 4-6% of all hospital discharge or admissions? And lines 44 and 45 for the wording, mortality rate ranging from approximately 2 to 5%. 

Response: Based on the description in the original cited literature [1], the 4-9% proportion specifically refers to the percentage of all hospital discharges among patients with diabetes mellitus as the primary cause for their acute hospital admission. Correction has been made in the revised manuscript (Page 4, Line 48-50).

3. Subjects and methods: The description of the type of the study should be mentioned. Please revise line 83 and figure 1 P21 for the exclusion criteria 1) age under 18 an over 75? Line 99 please revise the methodology of Cobas e601, Roche, Switzerland. It is electrochemiluminescence (ECL) technology for immunoassay analysis.

Response: Correction has been made in the revised manuscript (Page 5, Line 86-90; Page 6, Line 107; Page 7, Line 117; Fig 1).

Replies To Reviewer 2

1. There are severe limitations especially the limited sample size.

Response: Thank you for raising this critical issue. We acknowledge the limited sample size and have addressed this limitation in our original manuscript (Page 15-16, Line 295-299). Our strict inclusion and exclusion criteria, aimed at minimizing confounding factors affecting cortisol levels in predicting DKA-associated infections, resulted in a reduced sample size. Despite this limitation, our findings provide valuable insights into early prediction of bacterial infections in DKA patients. Future studies will focus on validating this predictive model with larger sample sizes.

2. Authors must specify the medications that affect the HPA axis in line 84.

Response: Correction has been made in the revised manuscript (Page 5, Line 91; Page 6, Line 92).

Replies To Journal Requirements

Response: The manuscript has been carefully revised to meet all of PLOS ONE's style requirements, including those for file naming. For your convenience, all modifications have been highlighted in red font to easily identify the changes.

2. Please ensure that you have an ORCID iD and that it is validated in Editorial Manager.

Response: I confirm that my ORCID iD has been validated in Editorial Manager.

Response: Thank you for pointing out the requirement for data availability in PLOS journals. We fully support the principles of data transparency and have taken steps to ensure compliance. We have included detailed data as supplementary information for easy access. We believe that this will facilitate reproducibility and further research based on our findings. Thank you again for your constructive input.

4. Please review your reference list to ensure that it is complete and correct. If you have cited papers that have been retracted, please include the rationale for doing so in the manuscript text, or remove these references and replace them with relevant current references.

Response: We have thoroughly reviewed the reference list and can confirm that it is complete and correct, with no citations to any retracted papers. The author information for reference 12, which was inadvertently omitted in the original submission, has been added in the revised manuscript. This addition has been highlighted in red for easy identification.

We hope that this further revised draft of the manuscript is now acceptable for publication in your journal and look forward to hearing from you soon. 

With best wishes,

Yours sincerely,

Liang Wang, Ph.D.

Corresponding author

E-mail: laiyangwangliang@163.com

References:

1. Dhatariya KK, Glaser NS, Codner E, Umpierrez GE. Diabetic ketoacidosis. Nat Rev Dis Primers. 2020;6(1):40.

---

## [Decision Letter · Decision Letter 1]

14 Jan 2025

Identification of early predictors and model for bacterial infection in diabetic ketoacidosis patients: A retrospective study

PONE-D-24-30892R1

Dear Dr. Wang,

We’re pleased to inform you that your manuscript has been judged scientifically suitable for publication and will be formally accepted for publication once it meets all outstanding technical requirements.

Kind regards,

Timotius Ivan Hariyanto, M.D.

Academic Editor

PLOS ONE

Additional Editor Comments (optional):

Reviewers' comments:

Reviewer's Responses to Questions

**Comments to the Author**

1. If the authors have adequately addressed your comments raised in a previous round of review and you feel that this manuscript is now acceptable for publication, you may indicate that here to bypass the “Comments to the Author” section, enter your conflict of interest statement in the “Confidential to Editor” section, and submit your "Accept" recommendation.

Reviewer #1: All comments have been addressed

Reviewer #2: All comments have been addressed

2. Is the manuscript technically sound, and do the data support the conclusions?

Reviewer #1: Yes

Reviewer #2: Yes

3. Has the statistical analysis been performed appropriately and rigorously? 

Reviewer #1: Yes

Reviewer #2: Yes

4. Have the authors made all data underlying the findings in their manuscript fully available?

Reviewer #1: Yes

Reviewer #2: Yes

5. Is the manuscript presented in an intelligible fashion and written in standard English?

Reviewer #1: Yes

Reviewer #2: Yes

6. Review Comments to the Author

Reviewer #1: The further revised draft of the manuscript is acceptable after fulfilling all the needed corrections.

Reviewer #2: A well conceptualised study with good methodology.

Manuscript is well-written

All comments have been addressed

7. PLOS authors have the option to publish the peer review history of their article (what does this mean?). If published, this will include your full peer review and any attached files.

Reviewer #1: **Yes: **Samia A. Girgis

Reviewer #2: No

---

## [Editor Report · Acceptance letter]

16 Jan 2025

PONE-D-24-30892R1 

PLOS ONE

Dear Dr. Wang, 

I'm pleased to inform you that your manuscript has been deemed suitable for publication in PLOS ONE. Congratulations! Your manuscript is now being handed over to our production team.

Kind regards, 

on behalf of

Dr. Timotius Ivan Hariyanto 

Academic Editor

PLOS ONE